# Placental Galectins in Cancer: Why We Should Pay More Attention

**DOI:** 10.3390/cells12030437

**Published:** 2023-01-29

**Authors:** Camille Fuselier, Alyssa Dumoulin, Alex Paré, Rita Nehmé, Samy Ajarrag, Philippine Granger Joly de Boissel, David Chatenet, Nicolas Doucet, Yves St-Pierre

**Affiliations:** INRS-Centre Armand-Frappier Santé Biotechnologie, Institut National de la Recherche Scientifique, 531 Boul. des Prairies, Laval, QC H7 V 1B7, Canada

**Keywords:** galectins, placenta, cancer

## Abstract

The first studies suggesting that abnormal expression of galectins is associated with cancer were published more than 30 years ago. Today, the role of galectins in cancer is relatively well established. We know that galectins play an active role in many types of cancer by regulating cell growth, conferring cell death resistance, or inducing local and systemic immunosuppression, allowing tumor cells to escape the host immune response. However, most of these studies have focused on very few galectins, most notably galectin-1 and galectin-3, and more recently, galectin-7 and galectin-9. Whether other galectins play a role in cancer remains unclear. This is particularly true for placental galectins, a subgroup that includes galectin-13, -14, and -16. The role of these galectins in placental development has been well described, and excellent reviews on their role during pregnancy have been published. At first sight, it was considered unlikely that placental galectins were involved in cancer. Yet, placentation and cancer progression share several cellular and molecular features, including cell invasion, immune tolerance and vascular remodeling. The development of new research tools and the concomitant increase in database repositories for high throughput gene expression data of normal and cancer tissues provide a new opportunity to examine the potential involvement of placental galectins in cancer. In this review, we discuss the possible roles of placental galectins in cancer progression and why they should be considered in cancer studies. We also address challenges associated with developing novel research tools to investigate their protumorigenic functions and design highly specific therapeutic drugs.

## 1. Introduction

Galectins (GAL) represent a family of evolutionarily conserved lectins that preferentially bind to β-galactose-containing glycoconjugates via their carbohydrate-recognition domains (CRDs), which consist of approximately 130 amino acids. Since 1994, galectins have been classified based on their CRD organization as a prototype, tandem-repeat type, or chimeric-type galectins [1]. In humans, 12 galectins have been identified, including GAL-1, -2, -7, -10, -13, -14, and 16 (prototypic), GAL-4, -8, -9, and -12 (tandem-repeat) and GAL-3 (chimeric type). In 2004, a landmark study on the ability of galectins to modulate outside-in signaling was published and attracted the interest of a broad audience, from glycobiologists to cellular and molecular biologists [2]. The study showed that the binding of galectins to cell surface glycoreceptors modulated the internalization of these receptors, thereby interfering with their signaling functions. Because alterations in the glycosylation pattern of cell surface receptors are a common feature of cancer cells [3], this study raised the hypothesis that glycan-binding proteins, such as galectins, play a decisive role in the fate of cancer cells.

Today, thousands of studies have been published on the roles of galectins in cancer. To summarize, galectins positively or negatively modulate tumor progression by exerting their extracellular activity at two primary levels. The first is based on their innate ability to regulate the immune response, either by neutralizing the cancer-killing function of immune cells [4,5] or neutralizing factors that attract immune cells to the tumor site [6]. Galectins can also contribute to cancer progression via other mechanisms, most notably by binding to soluble glycosylated immune mediators. A typical example is the ability of GAL-3 to neutralize the activity of interferon-gamma and other cytokines that promote the migration and infiltration of immune cells within the tumor [6,7]. Together, all these galectin-mediated functions establish galectins as promising targets for modulating cancer-specific immune responses, raising the interest of many working in immunotherapy. Indeed, this represents a very interesting avenue to improve immunotherapy success, as only 15–20% of patients achieve long-lasting results. [8,9,10,11].

The second protumorigenic role of galectins is their action on cell surface glycoreceptors expressed by cancer cells. This is accomplished via multiple intracellular mechanisms, such as conferring resistance to drug-induced cancer cell death, increasing the invasive behavior of cancer cells, inducing tumor-promoting genes, or simply increasing the proliferation rate of cancer cells [12]. A classic case is a study showing that the binding of GAL-1 to VEGF receptors can overcome the resistance of cancer cells to angiogenic inhibitors [13]. An important point to remember is that although a vast amount of literature has been published on the role of galectins in cancer, most of the studies have focused on a limited number of galectins. In particular, more than half of these studies have focused on GAL-3, and approximately another 25% have focused on GAL-1. There are many reasons justifying this interest in GAL-1 and -3, including that they were among the first identified and that most of the research tools available to study galectins, such as antibodies and genetically engineered cell and animal models, were developed for studies on these two galectins specifically. Only in recent years have we come to appreciate the role of other galectins in cancer, particularly GAL-7 and GAL-9 [14,15,16,17]. However, studies on the ability of other galectins, such as placental galectins, to regulate cancer progression remain somewhat limited, even though placental galectins share several functional features with more classical protumorigenic galectins.

Although considerable attention has been paid to the extracellular functions of galectins and their glycan-binding activity, galectins are also well known for their ability to promote cancer progression intracellularly [18]. This is not surprising as many galectins preferentially exist in cytosolic and nuclear compartments, consistent with the fact that they do not harbor a signal sequence and are exported outside the cells via a yet undefined non-classical mechanism [19]. In most cases, this intracellular activity involves carbohydrate-independent functions. Yet, this should not be so surprising. GAL-10 and GAL-16 have distinctive glycan binding sites (GBS) that preclude binding to β-galactoside [20,21]. For example, GAL-10, also known as the Charcot-Leyden crystals (CLC), binds in a carbohydrate-independent manner with intracellular RNases, modulating their translocation inside the eosinophils [22]. In the case of GAL-16, it binds via protein-protein interaction with c-Rel, a transcription factor known to play a central role in multiple types of cancer [23]. The ability of intracellular galectins to accomplish various functions via protein-protein interactions is not restricted, however, to galectins with non-functional GBS. This has been well established in the case of GAL-1 [24,25] and GAL-7, which binds BCl-2 [25,26,27]. In HeLa cells, disrupting the GAL-7/Bcl2 complex sensitizes cells to an apoptotic cell death [26]. Such carbohydrate-independent functions of galectins represent a paradigm shift in our comprehension of their biological activity, obliging us to rethink our strategies to inhibit their protumorigenic functions.

## 2. Placental Galectins and the Hallmarks of Cancer

The concept of placental galectins (also known as Chr19 cluster galectins) emerged many years ago in a landmark paper showing that GAL-13 (placental protein 13, or PP13), GAL-14, and GAL-16, all located within a cluster of genes on chromosome 19, were preferentially expressed in the placenta [28]. Except for GAL-16, which is stable as a monomer, GAL-13 and GAL-14 form homodimers (Figure 1).

Subsequent studies have shown that trophoblastic expression of these genes in the placenta is driven by DNA methylation and transcription factors known to regulate trophoblastic genes [31,32]. Consensus sequences are found within the promoter of placental galectins, such as those recognized by GATA, ESRRG (estrogen-related receptor gamma (ERR-gamma)), and the transcriptional enhancer factor TEF5 [33]. Today, the concept of placental galectins is well established. Their role in developmental processes and their potential use as biomarkers for gestational disorders have attracted the interest of many prenatal research investigators [33,34]. Many studies have since shown that other galectins, including those known to play a central role in cancer, such as GAL-1, -2, -3, -7 and -9, were also expressed at the maternal-fetal interface at different stages of pregnancy. In addition, abnormal expression patterns of galectins during pregnancy are associated with placental pathologies [35]. In contrast, there has been limited interest in studying whether GAL-13, -14, and -16 play a role in cancer progression even though cancer and pregnancy share many physiological properties detailed in many reviews over the past decade’s [36,37,38,39]. In fact, prenatal development shares several functional features recognized within the hallmarks of cancer as defined by Hanahan and Weinberg [40]. For example, they share common invasive mechanisms, most notably those regulating the invasiveness of the basement membrane and remodeling of the extracellular matrix (ECM). These processes involve proteolytic enzymes, such as matrix metalloproteinases (MMPs), a family of enzymes expressed by trophoblastic and cancer cells [36]. In cancer, the ability of galectins to induce MMPs to promote invasion is well established and has been shown to favor, for example, tumor growth and metastasis [41]. Such upregulation of MMPs in trophoblasts by placental galectins, including GAL-13 and -14, has also been reported recently [42,43]. Another common feature of cancer and human placental development is angiogenesis.

Vascular remodeling occurs in the maternal endometrium in preparation for embryo implantation and is essential for exchanging nutrients between the mother and fetus throughout pregnancy. The involvement of galectins in endothelial cell activation, proliferation and angiogenesis during pregnancy has been discussed in detail [44]. Similarly, the role of galectins in cancer angiogenesis for supplying nutrients and oxygen necessary for tumor growth has been recognized [45,46,47,48,49,50]. Although cancer angiogenesis is also considered a more chaotic process than normal and highly regulated angiogenesis [51,52], it provides cancer cells with the means to enter the vasculature and exit from the circulation to metastasize at secondary sites. The other central element shared by tumors and the placenta is the role of immune cells. In both cases, immune cells are essential for the induction of a local inflammatory environment that promotes cell clearance, angiogenesis and cell growth [39]. Galectins are also involved in the induction of immune tolerance. During pregnancy, this prevents the rejection of the fetus against an aggressive maternal immune response directed at non-self antigens (reviewed in [34]). In cancer, it promotes immune escape. In both cases, the immune-regulatory roles of galectins are essential [33,35,53]. Given these functional similarities, it is logical to conclude that placental galectins can potentially impact cancer progression via different mechanisms if expressed in cancer tissues.

## 3. Expression of Placental Galectins in Cancer Cells

Published studies hint that placental galectins are expressed in tissues other than the placenta and possibly in cancer tissues. In a report published in 1999 on the cloning of a cDNA encoding PP13 (GAL-13), Than et al. used a specific rabbit antiserum to show that GAL-13 was expressed at very high levels in the adult bladder (at even higher levels than in the placenta) and the spleen [54]. The authors also reported that PP13 was expressed in tumorous extracts from skin, brain, and liver carcinoma, as well as in fetal cells and various benign and malignant tumor tissues. To our knowledge, this is the first report indicating that placental galectins can be expressed in other tissues, including cancer tissues. More recently, our research team has used in silico and in vitro approaches to show that high expression of *LGALS14*, the gene encoding GAL-14, is associated with shorter survival in ovarian cancer cells [55]. We also found that *LGALS14* is preferentially expressed in high-grade serous adenocarcinoma (HGSA), the most aggressive subtype of ovarian cancer. Through in vitro studies of ovarian cancer cell lines, we further confirmed that *LGALS14* is readily expressed in HGSA. Using data from public databases, Kaminker and Timoshenko also showed that GAL-16 can be expressed in cells of lymphoid, epithelial, muscular and neuronal origins, albeit to lower levels than GAL-1, the most ubiquitously expressed member of the galectin family [56]. These results are consistent with the presence of several tissue-specific binding sites within the promoter of *LGALS16*. Overall, these studies support the hypothesis that placental galectins may be expressed in cancer cells.

## 4. Identification of Cancer Tissues Expressing Placental Galectins Using Public Databases

Database repositories of high-throughput gene expression data are a valuable resource to explore whether specific genes are expressed in other tissues and to guide future experimental research. Analysis of data pulled from these repositories confirmed that GAL-13, -14 and -16 have the strongest expression levels in the placenta (Figure 2A).

Interestingly, mRNA expression patterns of all three placental galectins were quite similar, and their expression in normal tissues was relatively low, especially for the *LGALS13* and *LGALS16* genes encoding human GAL-13 and GAL-16, respectively. This implies that targeting these galectins may generate minimal adverse side effects, which frequently account for the failure of drugs during clinical trials [57]. A notable exception is the brain, which has been shown to express all three galectins. The expression levels found in brain tissues confirm a recent report on *LGALS16* by Kaminker and Timoshenko (2021), who provided a detailed analysis of expression data in normal tissues and cell lines. The authors also observed *LGALS16* in several cancer tissues. To confirm that transcripts of all three placental galectins are expressed in multiple cancer tissues, we extended this analysis for the other two placental galectins (Figure 2B). Among other notable findings, we found higher *LGALS13* levels in lung and thyroid cancers compared to other cancer types. In the case of GAL-14, transcripts were found in many cancer tissues, most notably in cancers of epithelial origin. This includes gynecological cancers, such as ovarian cancer, a finding consistent with a previous [55]. As previously reported by Kaminker and Timoshenko (2021), transcripts encoding GAL-16 were found in many types of cancer, including breast, testicular, lung, and urothelial cancers. When the frequency of patients expressing *LGALS13* in their cancer tissues was analyzed, we found elevated levels of *LGALS13* in almost half of the patients with thyroid cancer. Additionally, approximately 60% and 20% of the thyroid cancer patients expressed *LGALS13* and *LGALS14,* respectively. In breast cancer, roughly 40% of patients expressed *LGALS13*, while very few expressed GAL-16 transcripts. However, for those GAL-16-positive patients, expression levels were significantly high. One of the most striking findings was that almost 75% of lung cancer patients expressed *LGALS14*. However, to our knowledge, no published studies have investigated the role of GAL-14 in lung cancer or its expression in lung cancer tissues.

## 5. Placental Galectin Expression Correlates with Cancer Progression

The above-cited studies and examination of public datasets provide strong indications of placental galectin expression in non-placental and cancer tissues. To evaluate whether the expression levels of these galectins affect tumor progression, we examined in public databases the association between expression levels of specific placental galectins and a positive or negative outcome in terms of survival. In the case of breast cancer patients, we found that high expression of any of the three placental galectins was associated with a statistically significant poorer outcome (Figure 3). Higher expression of placental galectins also correlated with poorer survival for patients with endometrial and ovarian cancer. In contrast, for thyroid cancer patients, high expression levels of placental galectins correlated with more prolonged survival. Such contrasting results are not surprising for galectins, given how they are well known for playing a dual role in cancer [18,48,58]. An excellent example of the contradictory roles of galectins is provided by our recent data showing that, while GAL-1 is associated with cancer progression, GAL-8 has a protective effect [59]. Additionally, although GAL-1 and GAL-7 are protumorigenic in colon cancer, GAL-4 acts as a tumor suppressor [60,61].

## 6. A Role for Placental Galectins in Cancer Progression?

Placental galectin expression in cancer cells does not necessarily mean they play a *de facto* role in cancer progression. Cancer cells express de novo many genes upon genetic alterations or epigenetic modifications. Such aberrant expression patterns often result from passenger mutations, defined as mutations that do not confer cancer cells with a selective growth advantage (in contrast to “driver mutations”) [63]. There are examples of galectin genes that are expressed de novo in cancer cells and that are capable of “driver” functions. This is not surprising considering the functional capabilities of galectins related to the hallmarks of cancer. A case in point is GAL-7, which has long been considered a skin-specific gene and a biomarker of stratified epithelia [64,65]. While GAL-7 is not detected in normal lymphoid cells, it is expressed in lymphoma [66]. Expression in lymphoma cells allows GAL-7 to accomplish driver functions by promoting the dissemination of tumor cells in peripheral organs through its ability to induce resistance to cell death and confer cancer cells with an invasive phenotype [15,67,68]. This aberrant expression of GAL-7 is likely triggered by DNA hypomethylation [69,70] or activation of signaling pathways associated with tumor progression, including gain-of-function mutations in the p53 gene [71,72]. Incidentally, the consequences of global or local hypomethylation, one of the hallmarks of cancer, on the expression of galectins are not restricted to GAL-7. It has also been reported in the cases of many, if not most, galectin family members, including GAL-1, -2, -3, -8, -9 and -12 [69,70,73,74]. Regarding GAL-1, which is also expressed in the placenta, the methylation status of its promoter regulates its expression in other tissues and nonplacental cell lines [75,76]. These studies support the idea that the expression of placental galectins may occur in cancer cells, most notably at advanced stages of tumors, in which genome-scale hypomethylation favors tumor progression [77].

## 7. Candidate Gene Pathways

Placental galectins were initially discovered as a trio of galectins commonly expressed in the placenta. As discussed above, this apparent inseparableness of the three seems to hold when expressed in other tissues. This is also observed, albeit to a lesser extent, in cancer cells’ chaotic gene expression patterns. Using the Human Protein Atlas database, it is possible to identify the nearest neighboring genes based on tissue RNA expression of all three placental galectins. We can see that placental galectins share a close functional relationship, as evidenced by the number of genes coexpressed with all three (Figure 4). Namely, these genes encode proteins, such as *ERVV-1*, *ERVV-2*, *NOTUM*, *KISS1*, *PWP1*, and *LIN28B*, which all have biological functions in pregnancy. For example, the *ERVV*-*1* and *ERVV-2* genes (also called *syncytins*) encode envelope glycoproteins that are highly expressed in normal placenta, specifically in the formation of placental syncytiotrophoblasts through cell-cell fusion [78]. Interestingly, the expression of the *syncytin-1* gene and its tissue specificity is controlled by DNA methylation of a CpG island in its proximal promoter. Although these functions are specific to human placental morphogenesis, hyper or hypomethylation of *syncytin-1* has been observed in several other tissues, notably in cancer cells arising from nonplacental tissues, such as the testis, ovary and colon [79,80]. A case in point is a recent study reporting that hypomethylation of *syncytin-1* gene promoter resulted in aberrant expression of the protein in testicular cancers, where GAL-13 is also constitutively expressed [81,82].

The *KISS1* gene encodes the neuropeptide kisspeptin, a well-established regulator of reproductive functions found in the syncytiotrophoblasts [83]. *KISS1* is highly expressed in the placenta and brain and weakly in the pancreas and kidney. However, kisspeptin is also highly expressed in the hypothalamic region and binds to its receptors encoded by *Kiss1R*, which is widely expressed in the brain. This suggests that the kisspeptin–KissR system may not only also be involved in non-reproductive functions [84]. Indeed, *KISS1* was initially discovered by the group of Danny Welch as a metastasis suppressor gene in the melanoma [85]. Interestingly, *KISS1* has been shown to play contradictory roles depending on the physiopathological conditions in which it is expressed. For instance, its expression in the brain is not favorable for patients during the development of glioblastoma, as *KISS1* has been shown to accelerate the metastatic ability of glioblastoma through the Gq-PLC-PKC pathway [79,86]. Moreover, the cellular process of KISS1 is related to several matrix MMPs in the brain and placenta [87]. These data raise the possibility that Syncytin-1 and GAL-13, and other galectins could positively or negatively impact cancer progression, possibly depending on the cell context.

The association of GAL-13 with *NOTUM* is interesting, as this gene encodes a palmitoyl-protein carboxylesterase that negatively regulates the Wnt signaling pathway by mediating depalmitoleoylation of Wnt proteins, rendering them inactive (Zhao et al., 2021). *NOTUM* knockdown in the HepG2 hepatoblastoma cell line has decreased migration and invasion while inhibiting tumor growth in vivo [88,89]. *NOTUM* is also involved in the colon adenocarcinoma [90,91].

In addition, we cannot ignore the association of GAL-13 with *LIN28B*, a member of the Lin-28 family. Members of this family are well known for their ability to control the Let-7 family of miRNAs biogenesis, which play a central role in the development and diseases [92]. High levels of LIN28A/LIN28B proteins are associated with many cancer biological behaviors and poor prognosis. Increased levels of LIN28B have pathological associations with many cancers, including colon cancer, lung cancer, hepatocellular carcinoma, ovarian cancer, germ cell tumors, prostate cancer, leukemia, breast cancer, oral squamous cell carcinoma, glioma and melanoma [93,94,95,96].

Additional information on the potential association between placental galectins and other cancer-related genes can be found in the literature. This is the case for *HOXA1*. This member of the homeobox (HOX) family of transcription factors plays an essential role during development. Mutations in *HOXA1* lead to several abnormalities, including neurological disorders and vascular malformations [97]. A recent study using a yeast two-hybrid system and pull-down experiments reported that GAL-13 binds to HOXA1 in HeLa cells and possibly regulates its function [98]. Interestingly, *HOXA1* expression is dysregulated in several types of cancer, such as gastric, prostate, breast and hepatocellular carcinoma, and promotes cancer progression [99,100].

Finally, the association between GAL-13 and the *PWP1* (periodic tryptophan protein-1) gene is of particular interest, as indicated by the use of Genemania. This web interface searches for functionally similar genes, helping to generate hypotheses about gene functions [101]. A search using *LGALS13* suggests a potential interaction between GAL-13 and *PWP1* gene, which encodes a protein found ubiquitously in several tissues, such as the testis, brain, endometrium, placenta, and thyroid. Although we still know very little about the role of this gene in cancer, two studies have shown that *PWP1* is upregulated in pancreatic cancer and may promote the progression of lung cancer [102,103].

## 8. Cautionary Notes

The above findings obtained through our search of the literature, public databases and the use of predictive bioinformatics tools pinpoint several hypothetical mechanisms of action and molecular pathways by which placental galectins could play a role in cancer progression. However, this review’s objective was not to conduct an extensive in silico analysis of the potential pathways through which placental galectins act in cancer progression. There are dozens of databases and algorithms available which were designed to facilitate investigations on the role of a given gene for a particular type of cancer, its prognostic value, its potential functional pathways, and the development of specific drugs. Thus, we leave it to the readers to adapt such in silico analyses for their needs and interest. Nonetheless, it is important to consider the limits of such in silico analyses in all cases. For example, most datasets are derived from high-throughput transcriptomic or proteomic experiments with cancer tissues. The datasets will thus not distinguish expression profiles in cancer cells from those in stromal cells. This will require further validation by various methods, such as immunohistochemistry profiling using antibodies, to confirm whether a specific placental galectin is expressed in cancer cells or a particular subset of stromal cells. It is also important to consider that observations of low expression levels in tissues might hide higher expression levels in a very small subset of cells. A case in point is GAL-7. Most transcriptomic data have shown that its expression level in breast cancer tissues is low. However, we subsequently showed that this protein is highly expressed in mammary basal epithelial cells [15]. Of course, the need for such validation steps generates another critical issue, namely the availability of specific research tools, which is especially important in the case of placental galectins. For instance, the cross-reactivity between anti-galectin antibodies is notorious, as galectins share relatively high primary, secondary, and tertiary structural homology [19]. This is particularly true for placental galectins (Figure 5A).

One also needs to be particularly prudent during investigations on the role of GAL-13 and GAL-16, which share the strongest homology among members of the galectin family (Figure 5B). Thus, unless antibodies have been tested for cross-reactivity between these two galectins, one needs to be very careful about interpreting any immunohistochemistry, western blot or ELISA data. This rings even truer if the antibodies or polyclonal antiserum were generated using the whole protein as an immunogen. Unfortunately, many suppliers do not provide this information in their specification sheets. This high homology among galectins has always been an obstacle to developing highly specific inhibitors or biomarkers. It will thus be important to generate novel and highly specific tools to study the role of human placental galectins in cancer.

## 9. Conclusions

To our knowledge, this is the first review addressing the potential role of placental galectins in cancer (Figure 6). Given their association with many cancer genes and their ability to modulate several hallmarks of cancer, there is a need to pay closer attention to placental galectins and gain more fundamental knowledge on their involvement in cancer progression.

## Figures and Tables

**Figure 1 cells-12-00437-f001:**
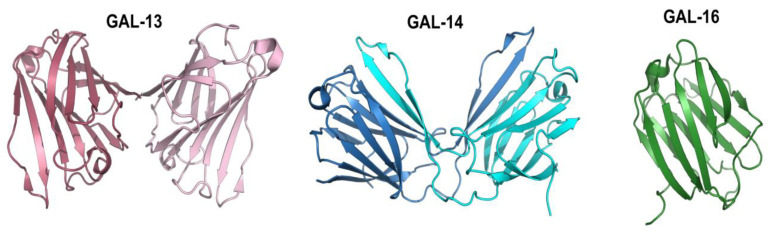
Three-dimensional structures of placental galectins. The dimeric structures of human GAL-13 and GAL-14 are shown respectively in purple/pink (GeneID UniProt Q9UHV8) and blue/cyan (GeneID UniProt Q8TCE9). GAL-13 is a prototype member stabilized by forming two disulfide bridges at the dimer interface [29]. In contrast, GAL-14 adopts a swapped dimer architecture, whereby terminal β-strands S5 and S6 of one monomer interact with the core structure of the opposite monomer (and vice versa) to form the canonical CRD ‘jelly-roll’ fold [30]. GAL-16 is shown in green (GeneID UniProt A8MUM7). Although prototype galectins typically crystallize as dimers, GAL-16 adopts a distinct monomeric structure [21].

**Figure 2 cells-12-00437-f002:**
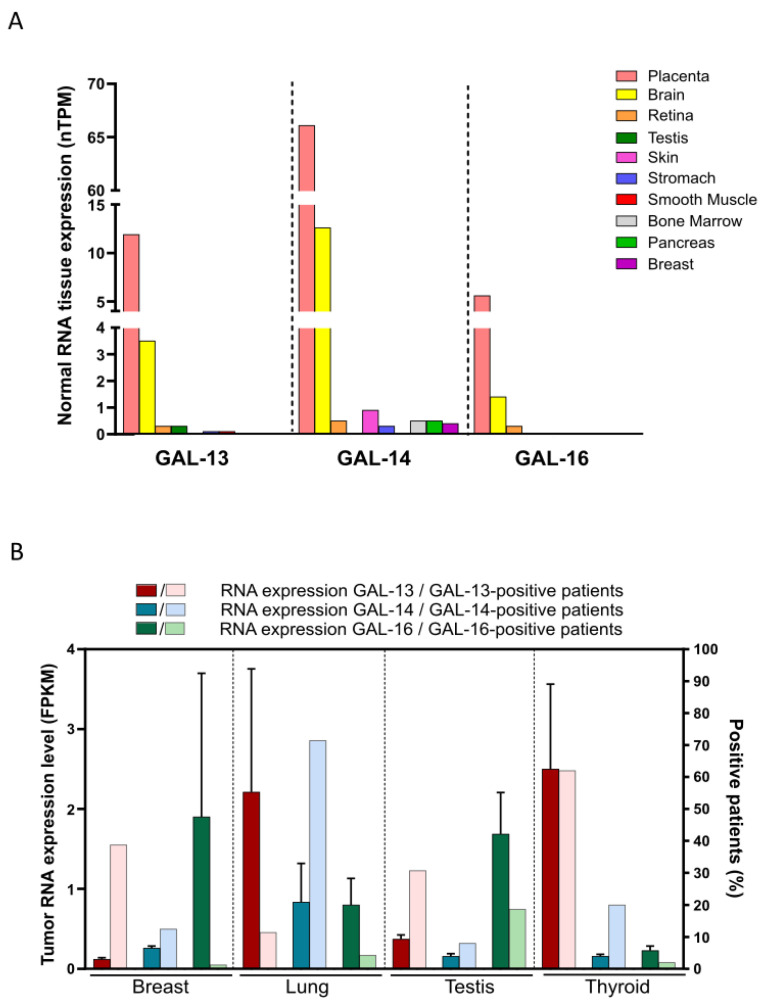
Comparison of placental galectins’ expression in normal and cancer tissues. (**A**) mRNA expression levels of placental galectins in normal tissues. Data were obtained from the Human Protein Atlas datasets. Normalized transcripts per million (nTPM) units are relative to normalized transcript expression values. (**B**) mRNA expression levels of placental galectins in cancer tissues. Data were obtained from the Human Protein Atlas datasets. FPKM units refer to fragments per kilobase of transcript per million mapped reads. The darker-colored histogram bars represent the mRNA expression levels of galectin, while the lightly colored histogram bars represent the percentage of patients with detectable expression levels.

**Figure 3 cells-12-00437-f003:**
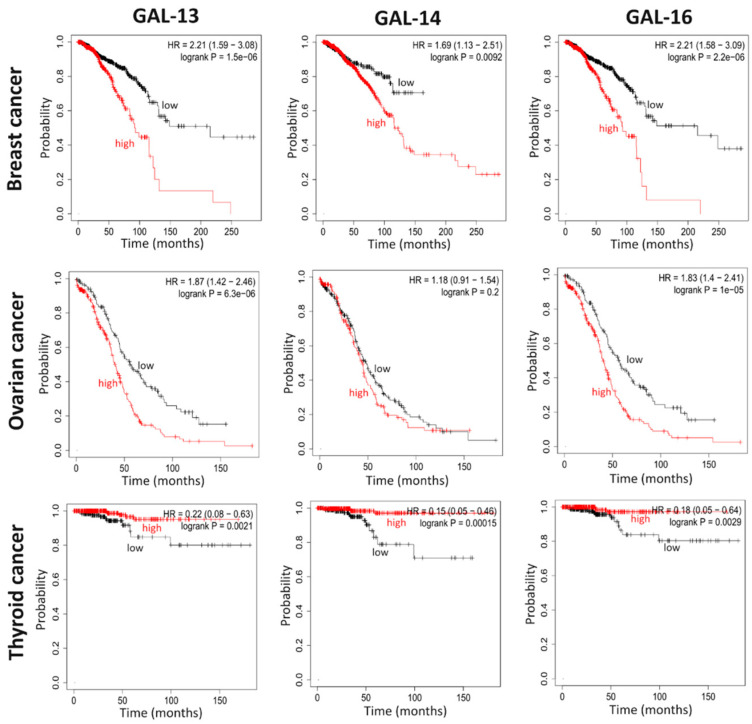
Kaplan-Meier survival curves of patients according to placental galectin levels. Data were extracted from Nagy et al. Pancancer survival analysis of cancer hallmark genes was performed using data from 2021 [62]. The log-rank test was used to detect significant differences between survival curves.

**Figure 4 cells-12-00437-f004:**
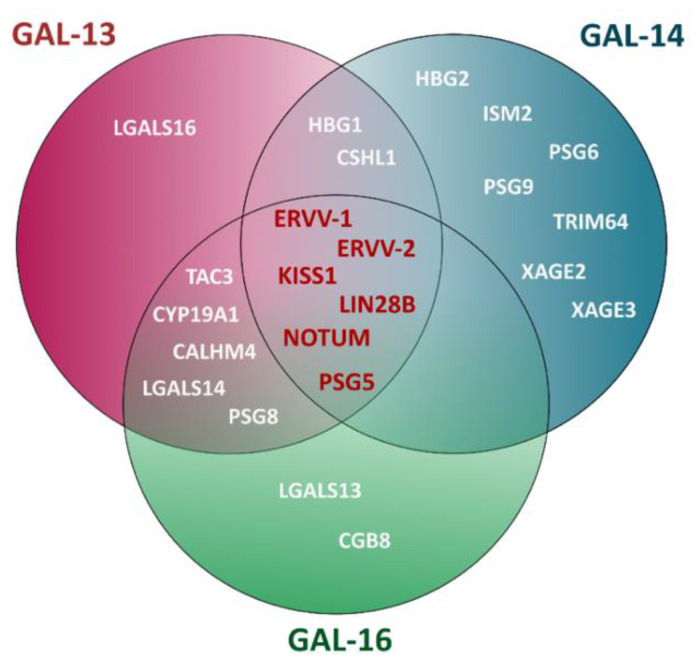
Venn diagram of genes co-expressed with placental galectins. The data were generated using the Human Protein Atlas datasets based on RNA-Seq expression data. The figure illustrates genes among the 15 closest neighboring genes associated with placental functions and cancer progression.

**Figure 5 cells-12-00437-f005:**
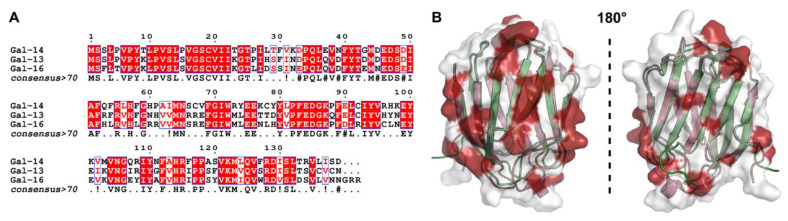
Structural and sequence homology between placental galectins. (**A**) Multiple sequence alignment between GAL-14, GAL-13, and GAL-16. The bottom consensus sequence was defined with a global score similarity threshold of 70%. Overall sequence identity is 68% between GAL-13 and GAL-14, 61% between GAL-14 and GAL-16, and 76% between GAL-13 and GAL-16. All sequences are numbered on top according to the consensus. Strictly conserved residues are highlighted in white font in red boxes. Conservation of residues Asn, Asp, Gln, Glu (#) and Ile, Val (!) are labeled in the consensus. The multiple sequence alignment was performed using Clustal Omega and visualized using ESPript 3.0. (**B**) Overlay between GAL-13 (purple cartoon) and GAL-16 (green cartoon) CRDs illustrates strong structural similarity (76% sequence identity). Electron density representation highlights surface positions that are conserved (white surface) or distinct (red surface) between both galectins. Since GAL-13 and GAL-16 share strong sequence homology, many residues form similar three-dimensional white surface epitopes that likely explain antibody cross-reactivity.

**Figure 6 cells-12-00437-f006:**
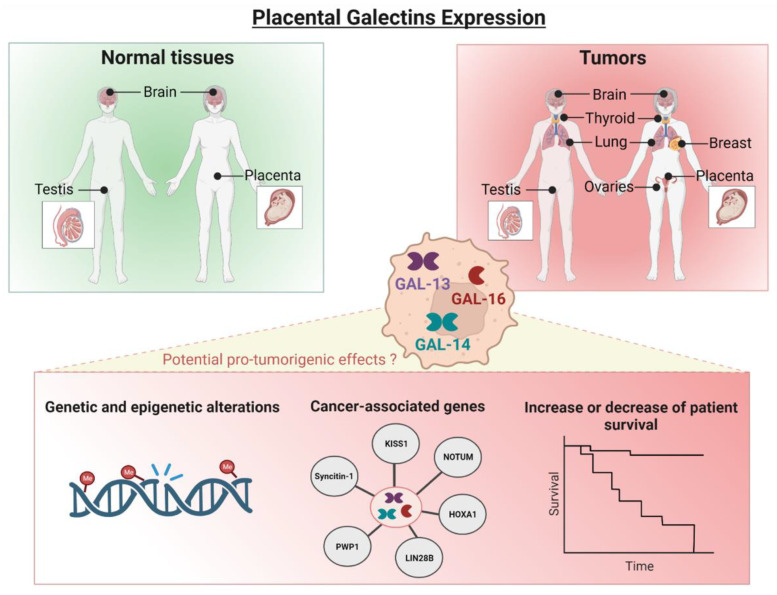
Recapitulative summary of the recent findings on placental galectins and potential future directions for further investigations.

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
