# Peer review of "Placental Galectins in Cancer: Why We Should Pay More Attention"

_cells, 2023, doi:10.3390/cells12030437_

Round 1
Reviewer 1 Report
This is an excellent review paper with a very clear concepts addressing the potential role of placental galectins in the context of cancer biology. There are few minor comments to be considered by the authors (both technical and some content):
11) Please edit and make sure to use proper fonts for human gene symbols, i.e. capitals and italic (lines 157-161 and more).
22) Figure 4: reference to Nagy et al is required (line 228).
33) Figure 5: it is not clear how these gene sets were obtained; please specify details of your search and methodology, i.e. how the ‘the nearest neighboring genes’ were recognized. Also, it is not clear why some genes you discussed in the section 7 (HOXA1, PWP1) are not appeared in this figure.
Author Response
Comments from reviewers:
Referee #1:
Comment: Lines 157-161 and more: Please edit and make sure to use proper fonts for human gene symbols, i.e. capitals and italic.
Reply: This has been corrected
Comment: Line 228: Reference to Nagy et al is required.
Reply: Corrected. The following reference has been added:
Nagy Á, Munkácsy G, GyÅ‘rffy B. Pancancer survival analysis of cancer hallmark genes. Sci Rep 2021; 11: 6047.
Comment: It is not clear how these gene sets were obtained; please specify details of your search and methodology, i.e. how the “the nearest neighboring genes” were recognized. Also, it is not clear why some genes you discussed in section 7 (HOXA1, PWP1) do not appear in this figure.
Reply: We have clarified this section by rearranging the order of appearance of the paragraphs and adding new information in the legend to figure 5. It is also important to note that figure 5 represents data generated by in silico analysis whereas HOXA1 and PWP1 have been identified by experimental systems.
Reviewer 2 Report
In this review paper, the authors discuss the potential roles of so-called "placental galectins" Gal-13, -14, and -16 in tumorigenesis.
The topic is intriguing and the authors provided a broad insight into the possible role of placental galectins in tumorigenesis and explained it clearly and argumentatively.
However, there are several shortcomings that I would like to point out, and I believe that the authors should further explain them.
Lines 48-52: In considering the role of galectins in tumorigenesis, the authors refer exclusively to the role of extracellular/membrane galectins, without mentioning the role of intracellular galectins. I believe that these roles should also be considered, or at least mentioned.
The authors provided many general arguments in favour of the possible role of placental galectins in tumorigenesis. Yet, they have not focused on or at least mentioned any of their specific roles in a particular biological process, e.g., cell proliferation or apoptosis that takes place during pregnancy, which are also crucial in tumorigenesis. They refer to the works of other authors, however, I believe that at least some of the examples should be mentioned in this paper.
In addition, there is one inconsistency in the summary itself:
Line 11: Today, the role of galectins in cancer is well established.
Line 16: Whether other galectins play a role in cancer remains unclear.
Please match.
Author Response
Referee #2:
Comment: Lines 48-52: In considering the role of galectins in tumorigenesis, the authors refer exclusively to the role of extracellular/membrane galectins, without mentioning the role of intracellular galectins. I believe that these roles should also be considered, or at least mentioned.
Reply: We have added a paragraph (lines 79 to 98) along with several references (including previous reviews on this topic) that describes in detail the intracellular role of galectins in cancer.
Comment: The authors provided many general arguments in favour of the possible role of placental galectins in tumorigenesis. Yet, they have not focused on or at least mentioned any of their specific roles in a particular biological process, e.g., cell proliferation or apoptosis that takes place during pregnancy, which is also crucial in tumorigenesis. They refer to the works of other authors; however, I believe that at least some of the examples should be mentioned in this paper.
Reply: We have added a sentence and cited a new review that describes in detail the roles of galectins during pregnancy. This manuscript reviewed in detail one of the fundamental role of galectins during pregnancy, i.e. its immunoregulatory role.
New reference: Balogh A, Toth E, Romero R, et al. Placental Galectins Are Key Players in Regulating the Maternal Adaptive Immune Response. Frontiers in Immunology; 10, https://www.frontiersin.org/articles/10.3389/fimmu.2019.01240 (2019, accessed 22 November 2022).
Comment: In addition, there is one inconsistency in the summary itself: Line 11: Today, the role of galectins in cancer is well established / Line 16: Whether other galectins play role in cancer remains unclear.
Reply: This has been corrected.